# Association of risk management practices and financial performance of microfinance institutions in Ethiopia: A two-step system generalized method of moments approach

**Megbaru Tesfaw Molla**[1]*, **Ratinder Kaur**[2]

**1** Department of Accounting and Finance, College of Business and Economics, Debre Markos University, Debre Markos, Ethiopia, **2** Department of School of Management Studies, Faculty of Business Studies, Punjabi University, Patiala, India

* tmegbaru@gmail.com, megbaru_tesfaw@dmu.edu.et

## Abstract

This study aims to analyze the relationship between risk management and performance of microfinance institutions (MFIs) in Ethiopia. An explanatory research design was employed by collecting secondary data from the financial statements of 24 sample MFIs from 2013/2014–2022/2024. The study utilized STATA version 15 software to address dynamic endogeneity bias, unobserved heteroskedasticity, and autocorrelation within individual units' errors in dynamic panel data using the generalized method of moments (GMM) estimator. According to the results of this study, both lagged variables, namely return on assets and return on equity, have a positive effect on the financing performance of Ethiopian MFIs. Financial leverage, interest rate risk, and cash-to-total deposits are negatively related to MFIs' returns on assets and return on equity. The relationship between loan loss provisions, loans and advances, and returns on assets is negative and significant but not with returns on equity. Loan-to-deposit ratios are positively related to financial performance indicators, but nonperforming loans are not significantly related to performance indicators. Thus, this study highlights strategies to enhance the financial performance and sustainability of Ethiopian MFIs. To do so, policymakers should foster an enabling environment for MFIs through financial literacy, risk management, and addressing financial challenges. In addition, MFI managers should focus on profitability, liquidity optimization, and effective credit evaluation to enhance performance.

## 1. Introduction

A microfinance institution (MFI) is a non-bank financial institution that offers loans, savings, insurance, and remittance services to low-income clients, especially to rural women [1]. As [2] argued, MFIs have come a long way since their modest

**Data availability statement:** All relevant data are within the manuscript and its Supporting Information files.

**Funding:** The author(s) received no specific funding for this work.

**Competing interests:** The authors assert that they do not have any conflicting interests.

**Abbreviations:** MFIs, microfinance institutions; GMM, generalize method of moments; ROA, return on asset; ROE, return on equity; ROAt-1, Lagged return on asset; ROEt-1, Lagged return on equity; NPL, nonperforming loan; LLP, loan loss provision; LA, loan and advances; FL, financial leverage; IRR, interest rate risk; CTD, cash to total deposit ratio; LTD, loan to deposit ratio; OE, operating expense ratio; ER, efficiency ratio (ER); FS, firm size; FGR, firm growth rate

beginnings, and some have even developed into world-class institutions that can compete with formal financial institutions. Since poverty remains a significant problem in the twenty-first century, MFIs are expected to grow and expand further [3]. MFIs are crucial for alleviating poverty and promoting societal peace [4]. However, Africa, particularly Ethiopia, faces limited access to financial services, which hinders long-lasting peace [5].

Ethiopia's National Bank has introduced microfinance legislation, allowing low-income households to access loans, micro-savings, insurance, money transfers, and leasing services as part of its economic development efforts, as per the MFIs Supervision Directorate. Following the adoption of the first microfinance legislation in 1996, a small group of MFIs was formed in early 1997 [6]. Ethiopia's MFIs are characterized by rapid growth, government-backed institutions, a rural focus, the promotion of credit and savings products, sustainability, and Ethiopian ownership. According to [7], the number of microfinance institutions increased by 43.4% and 13.8% in the previous year, with deposits reaching approximately 52.4 billion.

According to [8], risk management is crucial for microfinance institutions (MFIs) to reduce risk and align with their objectives, as reactive and poor strategies can lead to borrower default and failure to meet financial objectives. The most important risks for MFIs are credit, the market, and liquidity [9]. Ethiopian MFIs manage strategic, credit, liquidity, interest rates, and operational risks through risk identification, measurement, control, and monitoring. A sound risk management system includes active board oversight, adequate policies, and comprehensive internal control systems. Prior research has predominantly examined the relationship between risk management and the financial performance of banks, overlooking microfinance institutions (MFIs). Risk management is crucial for the sustainability and performance of MFIs, as poor strategies can lead to defaults and hinder financial goals. While most research focuses on banks, limited studies examine its impact on MFIs, especially in Ethiopia. This study aims to fill this gap by exploring the relationship between risk management practices and the financial performance of Ethiopian MFIs. Numerous studies have explored the association between MFI financial performance and credit risk [10–17]. However, MFI performance is influenced by factors beyond credit risk. Several researchers from various countries have investigated risk management practices and their relationships with MFI financial performance [18–21]. The findings of these studies have been mixed and inconsistent. Despite the importance of risk management for MFIs, particularly in Ethiopia, there is a lack of research on how various risk management practices influence their financial performance. Existing studies have mostly focused on credit risk, with mixed results, neglecting the broader range of risks that affect MFIs. This study seeks to fill this gap by examining the relationship between risk management practices and the financial performance of Ethiopian MFIs over a 10-year period. To address this gap, this study investigates the relationship between risk management practices and the financial performance of MFIs in Ethiopia. Data was collected from 24 sample MFIs over the period 2013/14–2022/23. This study contributes to the existing body of knowledge by focusing on the association of comprehensive risk management practices beyond just credit risk on the financial

performance of MFIs in Ethiopia. The objective of this study is to analyze the relationship between risk management and performance of microfinance institutions (MFIs) in Ethiopia.

## 1.1. Literature review

The current financial crisis has brought many banking and financial management issues to the forefront of academic discussion. In cases of financial failure, additional financial auditing is available. The Basel II recommendations support and strengthen the three pillars of achieving financial stability: regulations, supervision, and economic displacement, which affect business management based on sound business criteria. The Basel Accords draw on the Financial Conduct Authority's recommendations on financial supervision. The second (Basel II), first published in 2004, addressed three types of risk for banks. Market risk focuses on the uncertainty inherent in fluctuations in market prices. Credit risk is the risk of default by the counterparty of the contract. Operational risk refers to situations where something goes wrong. Business risk relates to part of our business proposition that is subject to uncertainty [14].

There are indications of the future of trade in India as early as 2000 BC; Investors diversify their investments by diversifying risks from the collective market, joint ventures to finance business trips. The Antwerp Stock Exchange was modeled after the Thomas Gresham Stock Exchange, founded in London in 1571 and later known as the Royal Stock Exchange. By the eighteenth century, risk trading activities flourished in England; Buyers and sellers enter into futures contracts to offset prices and delivery risk on products. In the 19th century, the development of modern risk management techniques went a step further with the creation of terminals, or futures exchanges, that allowed people to participate in exchanging the risk of various agricultural products. The United States was the country that established the first commodity exchange in New York and Chicago, due to the importance of the large agricultural sector to the economy. Similar projects were soon implemented in other major business centers such as London and Paris [15]. Microfinance has become a household name around the world as it is seen as a way to access services that are not available to large banks. The survival of MFIs in a country often depends on the general political and economic conditions of the country. However, the most important challenge faced by financial institutions globally in their financial intermediation role is how best to manage their loans, and financial institutions face increasing competition, pressure and economic and social risks, especially in developing countries [16]. Credit, interest rates, liquidity, and operational risks negatively affect MFIs' financial performance due to inadequate risk management by microfinance organizations [10]. Similarly, internal issues, external elements can impact the financial performance of MFIs, such as operating costs, over which the owners and the MFIs' chief executive officer have authority. According to the findings of this study, interest rate and financial leverage factors impact the financial performance of MFIs. Financial performance was shown to be positively and significantly influenced by interest rate risk and financial leverage risk [17]. [18] investigated the impact of credit risk based on non-performing loans on financial performance using empirical evidence from an emerging market of 200 banks across the Middle and East Africa region using panel data analysis using fixed and random effect. The result found that non-performing loan has a negative association with financial performance. According to [19] interest rates negatively impact returns on equity and assets, while the total debt-to-total assets ratio negatively affects returns on assets conducted by a study using a sample included 77 participants Rwandan financial institutions. Similarly, a study conducted [13] revealed a significant positive association between interest rates, credit risk, liquidity risk, and company value.

Capital adequacy, nonperforming loans, and management quality ratios significantly impact the return on assets of Nepali commercial banks, while the credit-to-deposit ratio and risk sensitivity do not [20]. [21] examined the association between credit risk management and financial performance using empirical data from Kenyan micro finance banks and found a significant negative correlation between ROA and equity performance measures, indicating a significant negative correlation between credit risk, portfolio risk, and loan loss provision coverage ratio parameters. Using random effects regression, [11] discovered that all three credit risk factors are strongly related to returns on assets and equity. Similarly, [22] propose that prudent credit risk had a positive effect on financial performance, except for loans and advances. [23]

determined the impact of risk management on the profitability of financial institutions in Sri Lanka from 2007 to 2011. The result of the study revealed that the financial and operating leverage of selected financial institutions positively influences profitability.

A study by [24] showed that liquidity risk negatively impacts financial performance, while credit risk significantly impacts financial performance when liquidity risk is present. As [25] founds, liquidity risk and the loan-to-deposit ratio negatively affect financial institutions' performance. [26] argued that liquidity risk significantly affects profitability, while nonperforming assets and nonperforming loans negatively affect profitability. [27] examined the influence of credit and operational risk on the financial performance of universal banks in Ghana using a structural equation model. Data were collected from 24 international banks in Ghana. The result revealed that operational and credit risks negatively influence financial performance.

[28] investigated risk management systems in Europe and focuses on France, Germany and the United Kingdom using 320 publicly traded companies. The results showed that effective risk management has a positive impact on the company's management and performance. While low debt negatively affects performance, environmental uncertainty also affects its quality. Similarly, [29] examined the relationship between financial risk and the performance of deposit banks on the stock exchanges of two selected West African countries using a sample of 20 bank deposits from 2009 to 2018. The result revealed that liquidity risk and credit risk have negative and significant effects on the performance of banks. The operational risk was discovered to have a positive and significant effect on the performance of Banks. [30] investigated the impact of risk management on financial performance of banks in Lebanon by collecting data from 39 commercial banks. The study found a direct relationship between market risk, liquidity risk, credit risk, and solvency risk and financial performance of banks. [31] examined the effect of credit risk management on the financial performance of commercial banks in South Asia. The findings suggest that leaders in South Asian countries should focus on improving capital adequacy to increase financial returns while reducing non-performing loans through routine procedures and strategies for non-performing loan risk management.

[32] studied the effect of credit risk management on financial presentation in Indonesian banking from 2017 to 2021. These study's findings shows that, the capital adequacy ratio has no intermediate impact on return on asset and return on equity, non-performing loan ratio has a negative and also significant influence on financial performance as assessed by return on asset and return on equity, loan to deposit ratio has no impact on financial presentation as assessed by return on asset and return on equity, net income margin gives no influence toward financial performance as calculated by the return on asset and return on equity. In other ways, [33] investigated the effect of market risk on the profitability of MFIs in Tanzania was assessed using interest rate and exchange rate risk as market risk measures and return on assets as profit measures. The study found interest rate has a positive impact on short-term results, and in the long run foreign exchange rate risk has a negative correlation with positive outcomes, while in the short run it has no significant impact on profitability. In their study, [34] evaluated the impact of liquidity risk on the performance of Islamic banks from 2012 to 2016. The outcome revealed a negative association between bank performance and liquidity measures. Similarly, [35] examined the efficacy of risk management practices that is liquidity risk and their impact on the financial performance of Islamic banks. The finding of this study revealed that the loan to total assets ratio cash to deposit ratio has a positive and significant impact on the return on asset and return on equity overall.

[36] used audited financial statements to evaluate the influence of operational risk on financial performance in the banking industry. The investigation revealed that the cost-to-income ratio has a significant negative effect on business performance, but the operating costs ratio has a significant positive effect on performance. Similarly, [37] also investigated the effects of operational risk management on the financial performance of commercial banks in Nigeria. The findings show a promising relationship between operational risk management and financial performance. [38] also examined the influence of operational risks and liquidity risks on the profitability of MFIs in Kenya. The findings revealed that operational risk and liquidity risk have significant negative profitability of micro finance banks in Kenya.

Ethiopia's MFIs are the most structured and rapidly growing in Sub-Saharan Africa, underpinned by Proclamation No. 40/1996. According to the report of [39] over six million borrowers are served with a loan portfolio of approximately USD 1.5 billion by Ethiopian MFIs. Ethiopian MFIs primarily use group-based lending and focus on rural poor populations, especially women and smallholder farmers. They maintain a relatively strong portfolio quality, with a PAR30 of 2.5%, though operational expenses are high (15–18%) due to rural outreach and infrastructure challenges. Their return on assets (ROA) stands at 1.5%, which lags behind peers in South Asia (2.5%) and Latin America (3–4%). This relatively low risk level in Ethiopia is partly due to conservative lending practices and strong group-lending methodologies that emphasize peer monitoring and social collateral. However, Ethiopian MFIs also operate under tight regulatory constraints and a limited technological environment, which restricts their ability to diversify products and manage risk dynamically. Ethiopian MFIs have expanded outreach to rural communities but offer small loans (USD 250–300) and limited services beyond credit. In comparison, South Asian MFIs provide larger loans and financial literacy programs, while Latin American MFIs focus on individual lending in urban areas [40].

Ethiopian MFIs manage credit risk through strict client screening and group lending, while South Asian MFIs add financial education and Latin American MFIs use credit scoring and diverse products for higher returns. Operational risks in Ethiopia stem from weak digital infrastructure and manual processes, unlike Kenya and India, where mobile tech and MIS reduce costs and enhance efficiency. Liquidity risk is high in Ethiopia due to reliance on donor funding, unlike Latin America's diversified funding sources, which improve resilience [41].

## 1.2. Research gap

A review of the related literature presented above indicates that risk management plays an important role in the success and continuity of MFIs. Despite all the available related research, the association of risk management practices, and the financial performance of MFIs in Ethiopia have not been adequately examined. According to the examination of the literature, most of the research studies have conducted on commercial banks' risk management practices, with little emphasis paid to MFIs. Most researchers across different countries have conducted their studies on the relationship with the financial performance of MFIs in various nations, ignoring Ethiopia. Despite this, Ethiopia is affected by international country-specific factors, such as economic conditions, institutional computation, financial regulations, and business principles. Hence, this study helps to fill the gap in the literature on risk management practices of MFIs. Thus, the objective of this study is to analyze the association of risk management practices and the financial performance of MFIs in Ethiopia.

## 1.3. Hypotheses of the study

The following research hypotheses are developed parallel to the literature review:

H1: Nonperforming loans have a negative association with the financial performance.

H2: Loan loss provisions are negatively associated with the financial performance.

H3: Loans and advances have a negative association with the financial performance.

H4: Financial leverage is negatively associated with the financial performance.

H5: Interest rate risk is negatively associated with the financial performance.

H6: The cash-to-total deposit ratio has a negative association with the financial performance.

H7: The loan-to-deposit ratio has a negative association with the financial performance.

H8: The operating expense ratio is negatively associated with the financial performance.

H9: The efficiency ratio is negatively associated with the financial performance.

### 1.4. Conceptual framework of the study

According to prior studies [15–17,25], the following conceptual framework was developed (refer Fig 1):

## 2. Research methodology

### 2.1. Study approach and design

A quantitative research approach is used in this study. This study used an explanatory research design. The study collected secondary data from audited financial statements of Ethiopian MFIs that addressed the study's purpose.

### 2.2. Sources of data and method of data collection

This study selected a sample of MFIs using a nonprobability sampling method. Currently, there are 34 MFIs, which are considered as the study population, are operating and registered as AEMFIs in Ethiopia [35]. In this study, using purposive sampling, 24 MFIs were selected from 34 MFIs in Ethiopia based on their age since they incorporated and availability of audited financial statements for the study period of 2013/14–2022/23. Accordingly, 10-year-old MFIs were considered as a sample of the study. The justification for this is that MFIs that were established before 2013/14 are larger, which means that their risk management departments have more staff and are well structured, which is the focus of this study. Therefore, 24 MFIs established before 2013/14 were selected for this study.

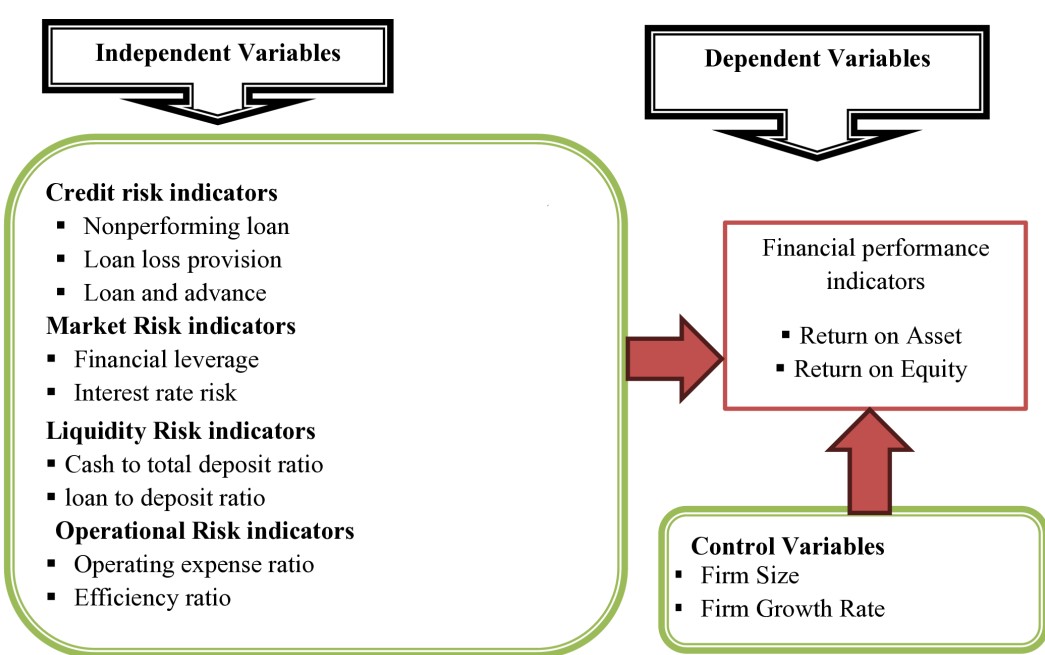

Source: Author's compilation (2023)

**Fig 1. Conceptual framework of the study.**

To assess the reliability of the cross-sectional sample, a statistical power analysis was conducted based on a sample of 24 microfinance institutions (MFIs), incorporating 10 predictors—including lagged ROA/ROE, non-performing loans, loan loss provisions, loans and advances, interest rate risk, the cash-to-total deposit ratio, the loan-to-deposit ratio, operating expenses, and the efficiency ratio. Assuming a medium effect size ($f^2 = 0.15$) and a significance level of $\alpha = 0.05$, the resulting statistical power was approximately 5.37%, which falls substantially below the conventional benchmark of 80%. This suggests a limited capacity to detect moderate relationships within the cross-sectional framework. To mitigate this limitation, the study employed a dynamic panel data model using the Generalized Method of Moments (GMM) over a ten-year period. This longitudinal approach leverages within-entity variation and enhances the robustness of the analysis.

To ensure the reliability and validity of the data used in this study, several methods were employed. First, purposive sampling was used to select MFIs established before 2013/14, as they are larger, more structured, and have better-developed risk management departments, ensuring consistency and relevance to the study's focus and enhancing data reliability. Second, the study included only MFIs with audited financial statements available for the entire study period (2013/14–2022/23), as such statements enhance accuracy and credibility due to their independent verification. Third, the study period was clearly defined as ten years, providing consistency in data collection and analysis while offering a robust understanding of trends and minimizing short-term fluctuations that could undermine data reliability. Fourth, the sample of 24 MFIs was carefully chosen from the total population of 34 MFIs registered under AEMFIs in Ethiopia. By focusing on institutions operational for at least ten years, the study ensured the representation of MFIs with sufficient data, thereby supporting the validity of the findings. Lastly, the inclusion of older and larger MFIs allowed the study to concentrate on institutions with well-structured and mature risk management practices, which increased the relevance of the findings and reduced potential variability. By implementing these measures, the study ensured that the data collected was both reliable and valid, thus supporting the credibility of its findings and conclusions.

## 2.3. Method of data analysis and model specification

The study utilized secondary data analysis, beginning with descriptive analysis to summarize dataset characteristics and pairwise correlation analysis to evaluate the association between dependent and independent variables. Regression analysis was conducted using the Generalized Method of Moments (GMM) to explore the core relationships between variables, with results presented in tables. GMM was chosen due to its effectiveness in handling endogenous variables, particularly lagged dependent variables, in short-term panel data with a large number of observations (N) and a small number of time periods (T). The study analyzed 10 years of data with 240 panel observations. Traditional fixed and random effects models commonly used in financial industry research were deemed unsuitable due to challenges arising when lagged dependent variables are significant over multiple periods. When working with short-term panel data, the generalized method of moments (GMM) becomes a helpful tool for investigating the relationship between variables [42]. The GMM was employed in this study due to its suitability for addressing key challenges inherent in short-term panel data analysis. Specifically, GMM is advantageous in models containing endogenous variables, as it effectively mitigates issues of endogeneity and unobserved heterogeneity by using appropriate instruments. Arellano and Bond created dynamic panel data (DPD) models to address the unobserved heterogeneity often encountered with panel data. To minimize this problem, [43] developed different GMM models by differencing all repressors and employing GMM. The usual approach when facing heteroskedasticity of unknown form today is to use the GMM introduced by [42]. The Arellano–Bond approach estimates large N panels, a small T, a few periods, and many units. According to [42] "a two-step GMM estimate tends to have smaller variance than a one-step GMM estimate." Furthermore, according to [44], two-step GMM estimators have asymptotic advantages over one-step estimators. In this study, STATA Version 15 was used for data analysis using a two-step GMM estimator. STATA was chosen for its robust statistical tools in panel data analysis, regression modeling, and hypothesis testing [45]. This software is efficient in handling longitudinal data ensures accurate results, while

advanced data management, user-friendly features, and high-quality visualizations enhance transparency and interpretability. Thus, the study used STATA as it widely accepted for analyzing the financial performance and risk management practices of Ethiopian MFIs. This study used the following general econometric model, which is similar to that employed by [30,37,38,40,41,46].

$$Y_{i,t} = \beta o + \partial Y_{i,t-1} + \sum \beta i X_{i,t} + \mu_i + \varepsilon_{i,t}$$

where: $Y_{i,t}$ = MFI's i's performance in year t, namely, ROA$_{i,t}$ and ROA$_{i,t}$, which are the return on assets and return on equity, respectively, β0 = constant term of the model, $Y_{i',t-1}$ = MFI' i's performance in year t-1, $X_{i,t}$ = a vector of current values of MFI's specific explanatory variables, $\mu_i$ = the specific time-invariant effect of unobserved MFI., $\varepsilon_{i,t}$ is the error term of the model.

When the above general model is modified to include the variables in this study, the regression equations are as follows:

$$ROA_{i,t} = \beta0 + \beta1(ROA_{i,t-1}) + \beta2(NPL_{i,t}) + \beta3(LLP_{i,t}) + \beta4(LA_{i,t}) + \beta5(FL_{i,t}) + \beta6(IRR_{i,t})$$
$$+\beta7(CTD_{i,t}) + \beta8(LTD_{i,t}) + \beta9(OE_{i,t}) + \beta10(ER_{i,t}) + \beta11(logFS_{i,t}) + \beta12(FGR_{i,t}) + \varepsilon_{i,t} \quad (1)$$

$$ROE_{i,t} = \beta0 + \beta1(ROE_{i,t-1}) + \beta2(NPL_{i,t}) + \beta3(LLP_{i,t}) + \beta4(LA_{i,t}) + \beta5(FL_{i,t}) + \beta6(IRR_{i,t})$$
$$+\beta7(CTD_{i,t}) + \beta8(LTD_{i,t}) + \beta9(OE_{i,t}) + \beta10(ER_{i,t}) + \beta11(logFS_{i,t}) + \beta12(FGR_{i,t}) + \varepsilon_{i,t} \quad (2)$$

where: ROA$_{i,t}$ is MFIi Return on Asset at time t, ROA$_{i,t-1 \text{ is the}}$ lagged dependent variable of ROA$_{i,t}$, ROE$_{i,t}$ is the MFIi return on equity at time t, ROE$_{i,t-1 \text{ is the}}$ lagged dependent variable of ROE$_{i,t}$, NPL$_{i,t}$ is the nonperforming loan for MFIi at time t, LLP$_{i,t}$ is the loan loss provision for MFIi at time t, LA$_{i,t}$ is the loan and advances for MFIi at time t, FL$_{i,t}$ is the financial leverage for MFIi at time t, IRR$_{i,t}$ is the interest rate risk for MFIi at time t, CTD$_{i,t}$ is the cash-to-total deposit ratio for MFIi at time t., LTD$_{i,t}$ is the loan-to-deposit ratio for MFIi at time t, OE$_{i,t}$ is the operating expense ratio for MFI i at time t, ER$_{i,t}$ is the efficiency ratio for MFIi at time t, logFS$_{i,t}$ is firm size and is the natural logarithm of total assets for MFIi at time t., FGR$_{i,t}$ is the firm growth rate: change in annual revenue for MFIi at time t., $\varepsilon_{i,t}$ is the error term of the model for MFIi at time t.

### 2.4. Variables definition and measurements

1. **Financial Performance Indicators:** The common profitability measures used to gauge the financial performance of financial institutions are return on assets (ROA) and return on equity (ROE) [10,12,19–21,44]. The return on assets indicates the profitability of the firm's assets after all expenses and taxes are deducted [17]. The return on equity indicates the firm's profitability to shareholders after all expenses and taxes are deducted [47].

2. **Risk Management Indicators**

A. **Credit Risk Indicators**

  A. **Nonperforming Loan (NPL).** A nonperforming loan is a loan in which borrowers' default because of the nonpayment of the loan [48]. Because of this, the profitability of the bank can be negatively affected [49]. Lower interest income and higher loan loss provisions result from poor asset quality. Over time, high nonperforming loans can lead to greater bank risk [50].

  B. **Loan loss provision (LLP).** Accountants and financial economists have long held concerns that inefficient loan loss accounting may have a material impact on reported capital and earnings, especially in the banking industry [51]. Financial institutions create a loan loss provision to set aside funds for default or risk.

C. **Loan and Advances (LA).** The coefficient of the ratio of loans to advances has the most significant positive effect on profitability across financial institutions [52]. If the ratio of loans and advances is high, financial institutions might not have enough liquidity to cover unforeseen funding requirements or economic crises.

B. Market Risk Indicators

A. **Financial Leverage (FL).** Increasing financial leverage can benefit firm performance if the balance of debt and equity does not exceed each other. Financial leverage measures the degree of financial risk. Therefore, the higher the ratio is, the riskier the financial institutions are [53]. [54] found that financial leverage can positively influence firm performance because it can be treated as a tool for disciplining management, which is expected on the basis of agency cost theory.

B. **Interest rate risk (IRR).** The net interest margin, sometimes called the spread margin, can monitor interest rate risk. After interest is paid on all liabilities, this ratio calculates how much income the institution still has left after interest is paid and compares it to either the institution's total assets or its performing assets [55]. This means that interest rates play a key role in influencing the financial performance of an organization ([17].

C. Liquidity Risk Indicators

A. **Cash-to-total deposit ratio (CTD).** The cash-to-total deposit ratio is how much a financial institution lends out of its mobilized deposits. This variable indicates how much of a financial institution's core funds are used for lending, the main banking activity. There is a direct relationship between the cash-to-deposit ratio and the financial performance of financial institutions lending money to borrowers [56].

B. **Loan-to-Deposit Ratio (LTD).** [57] investigated the impact of liquidity management on the performance of deposit money banks. The empirical analysis shows that liquidity management significantly impacts deposit money banks' performance in Nigeria. For the liquidity and cash reserve ratios, the loan-to-deposit ratio has a negative effect. Nevertheless, the key results showed that financial institutions with optimal liquidity maximize their returns.

D. **Operational Risk Indicator**

A. **Operating Expense Ratio (OE):** The operating expense ratio provides a general overview of how efficiently the MFI uses assets. The operating expense ratio recommended monitoring tools for inefficiency risk. The operating expense ratio is calculated by dividing total expenses by assets ([17]. The operating expense ratio is one of the efficiency ratios used to gauge an organization's efficiency [58].

B. **Efficiency Ratio (ER):** This ratio analyzes the extent to which total income (net interest and other income before loan loss provision) is consumed by expenses. This helps MFIs focus not only on how costs affect efficiency but also on the impact of revenue. This ratio shows how many dollars are earned per dollar spent [58].

3. **Control Variables:** In addition to the independent variables, Firm Size (FS) and Firm Growth Rate (FGR) were included in this study as control variables (Table 1).

## 3. Results and discussions

Before running the regression, the data set was checked for panel data diagnostic tests that researchers required to examine the data set for the analysis results to be reliable and valid. Based on this, the researchers conducted normality tests and multicollinearity, and the data set found no normality or multicollinearity problems. Finally, the heteroskedasticity test was checked, and the data set found a problem of heteroskedasticity. Thus, the null hypothesis was rejected. Therefore, the GMM model has been used to minimize heteroscedasticity by incorporating lagged dependent variables of return on assets ($ROA_{t-1}$) and return on equity ($ROE_{t-1}$).

**Table 1. Summary of Variables Measurements.**

| Dependent Variables | Variable | Label | Measurement |
|---|---|---|---|
| Financial Performance indicators | Return on Asset | ROA | Net income after tax/Total assets |
| | Return on Equity | ROE | Net income after tax/Total Equity |
| **Independent Variables** | **Proxies** | | **Measurement** |
| **Credit risk indicators** | Nonperforming Loan | NPL | Nonperforming Loans/Total Loans and advances |
| | Loan loss provision | LLP | Loan loss provision/Gross Loan |
| | Loan and Advances | LA | Total loan and advances/Total Deposits |
| **Market Risk indicators** | Financial Leverage | FL | Earnings before interest and tax/Earnings before interest and tax-Interest |
| | Interest Rate Risk | IRR | Interest income-interest expense/Total assets. This is called net interest margin |
| **Liquidity Risk indicators** | Cash to Total Deposit Ratio | CTD | Total cash/total deposit |
| | Loan to Deposit Ratio | LTD | Total loan/total deposit |
| **Operational Risk indicator** | Operating Expense Ratio | OE | Total operating expense/Total asset |
| | Efficiency Ratio | ER | Total expense before tax/net interest income before provision + other income |
| **Control Variables** | Firm Size | FS | Natural logarithm of total assets of the firm |
| | Firm Growth Rate | FGR | Change in the firm's annual revenue that is $[(revenue_t - revenue_{t-1})/revenue_{t-1}]$ |

Source: Author's compilation (2023).

## 3.1. Descriptive statistics analysis

This study examined fourteen variables among 24 Ethiopian MFIs over ten years, resulting in 242 observations and descriptive statistics (Table 2).

The mean value of return on assets is 0.162%, with a standard deviation of 0.259%. The lagged return on assets (ROAt-1) has a mean value of 0.169 and a standard deviation of 0.249. The average return on equity is 0.412%, with a deviation of 0.576%. The return on equity is ranked first in terms of mean and maximum values. This study measures risk

**Table 2. Descriptive Statistics of Variables.**

| Variable | Obs | Mean | Std. Dev. | Min | Max |
|---|---|---|---|---|---|
| ROA | 240 | .162 | .259 | 0 | 1.57 |
| ROA$_{t-1}$ | 240 | .169 | .249 | 0 | 1.57 |
| ROE | 240 | .412 | .576 | 0 | 3.979 |
| ROE$_{t-1}$ | 240 | .421 | .564 | .002 | 3.979 |
| NPL | 240 | .006 | .018 | 0 | .145 |
| LLP | 240 | .008 | .019 | 0 | .145 |
| LA | 240 | 2.233 | .842 | .491 | 4.493 |
| FL | 240 | 1.564 | 1.236 | −6.29 | 7.568 |
| IRR | 240 | .113 | .049 | .017 | .19 |
| CTD | 240 | .671 | .798 | .026 | 7.17 |
| LTD | 240 | 2.166 | 1.308 | .034 | 15.992 |
| OE | 240 | .153 | .114 | .013 | .761 |
| logFS | 240 | 5.306 | .992 | 2.796 | 7.59 |
| FGR | 240 | .298 | 1.079 | −.999 | 12.218 |

Note: ROA = Return on asset, ROE = return on equity, ROA$_{t-1}$ = lagged return on assets, ROE$_{t-1}$ = lagged return on equity, NPL = nonperforming loan, LLP = loan loss provision, LA = loan and advances, FL = financial leverage, IRR = interest rate risk, CTD = cash to total deposit ratio, LTD = loan to deposit ratio, OE = operating expense ratio, ER = efficiency ratio, FS = firm size, FGR = firm growth rate.

Source: Researchers' computations 2023.

management practices for multifunctional institutions (MFIs) in Ethiopia using various measures. The average value of nonperforming loans is 0.006, with 0.018 variations. The loan loss provisions are 0.008, with 0.019 variations. The values of Loans and Advances are 0.491 and 4.493, respectively. The financial leverage is 1.564, with values of −6.29 and 7.568, respectively. The interest rate risk has standard deviations of .113 and 0.046, with maximum and minimum values of 0.017 and 0.19, respectively. The study period's average cash-to-total deposit ratio is 0.671, with a standard deviation of .0798. The mean loan-to-deposit ratios are 2.166 and 1.308, with maximums of 0.034 and 15.992, respectively. The operating expense ratio has a mean of 0.153, and the efficiency ratios have means of 0.789 and 0.492. This study analyzes descriptive statistics and control variables for MFIs in Ethiopia.

### 3.1. Correlation analysis and its implications

According to Table 3 below, according to the Pearson correlation, the dependent variable, ROA, and lagged dependent variable are positively and significantly correlated at the 1% significance level. Loan and advance, financial leverage, interest rate risk, the loan-to-deposit ratio, and the log of firm size are negatively and significantly correlated at the 10%, 1%, 1%, 5%, and 1% significance levels with return on assets, respectively. However, nonperforming loans, loan loss provisions, the cash-to-total deposit ratio, the operating expense ratio, and the firm growth rate are not significantly correlated with returns on assets.

The correlation coefficients in Table 3 below are 72.8%, −4.3%, −0.4%, −11.9%, −22.6%, −33.5%, 1.6%, −13.9%, 8.7%, −57.6%, and −6.8% for the nonperforming loan, the loan loss provision, the loan and advances, financial leverage, interest rate risk, and cash-to-total deposit, loan-to-deposit ratio, operating expense ratio, efficiency ratio, firm size, and firm growth rate, respectively.

According to Table 4 below, according to the Pearson correlation, the dependent variable, ROE, and lagged dependent variable are positively and significantly correlated at the 1% significance level. Loan and advance and financial leverage, interest rate risk, loan-to-deposit ratio, operating expense ratio, log of firm size, and firm growth rate are negatively and significantly correlated with returns on assets at the 5%, 1%, 1%, 10%, 5%, 1% and 5% levels of significance, respectively. However, nonperforming loans, loan loss provisions, and the cash-to-total deposit ratio are not significantly correlated with returns on assets.

**Table 3. Pairwise Correlation Analysis: ROA as a firm financial performance proxy.**

| Variables | ROA | ROAt1 | NPL | LLP | LA | FL | IRR | CTD | LTD | OE | logFS | FGR |
|---|---|---|---|---|---|---|---|---|---|---|---|---|
| ROA | 1.000 | | | | | | | | | | | |
| ROA$_{t-1}$ | 0.728*** | 1.000 | | | | | | | | | | |
| NPL | −0.043 | −0.069 | 1.000 | | | | | | | | | |
| LLP | −0.004 | 0.051 | 0.720*** | 1.000 | | | | | | | | |
| LA | −0.119* | −0.120* | 0.064 | 0.022 | 1.000 | | | | | | | |
| FL | −0.226*** | −0.223*** | −0.014 | −0.121* | −0.002 | 1.000 | | | | | | |
| IRR | −0.335*** | −0.266*** | 0.027 | 0.063 | 0.360*** | 0.061 | 1.000 | | | | | |
| CTD | 0.016 | 0.175*** | −0.012 | 0.002 | −0.079 | −0.103* | −0.320*** | 1.000 | | | | |
| LTD | −0.139** | −0.090 | 0.064 | 0.100 | 0.415*** | −0.041 | 0.103 | 0.382*** | 1.000 | | | |
| OE | 0.087 | 0.201*** | 0.029 | 0.124* | 0.023 | −0.058 | 0.062 | −0.036 | −0.036 | 1.000 | | |
| logFS | −0.576*** | −0.549*** | −0.091 | −0.236*** | −0.008 | 0.140** | 0.196*** | −0.168*** | −0.012 | −0.473*** | 1.000 | |
| FGR | −0.068 | 0.085 | 0.029 | 0.016 | −0.079 | −0.007 | 0.050 | −0.003 | 0.007 | −0.096 | 0.083 | 1.000 |

Note: ROA = Return on asset, ROA$_{t-1}$ = lagged return on assets, NPL = nonperforming loan, LLP = loan loss provision, LA = loan and advances, FL = financial leverage, IRR = interest rate risk, CTD = cash to total deposit ratio, LTD = loan to deposit ratio, OE = operating expense ratio, ER = efficiency ratio, FS = firm size, FGR = firm growth rate. ***, **, and * indicate significance levels less than 1%, 5% and 10%, respectively.

Source: Researchers' computations.

**Table 4. Pairwise Correlation Analysis: ROE as a firm financial performance proxy.**

| Variables | ROE | ROEt1 | NPL | LLP | LA | FL | IRR | CTD | LTD | OE | logFS | FGR |
|---|---|---|---|---|---|---|---|---|---|---|---|---|
| ROE | 1.000 | | | | | | | | | | | |
| ROEt1 | 0.609*** | 1.000 | | | | | | | | | | |
| NPL | −0.049 | −0.069 | 1.000 | | | | | | | | | |
| LLP | 0.010 | 0.014 | 0.604*** | 1.000 | | | | | | | | |
| LA | −0.128** | −0.072 | 0.070 | 0.049 | 1.000 | | | | | | | |
| FL | −0.228*** | −0.220*** | 0.000 | −0.121* | −0.065 | 1.000 | | | | | | |
| IRR | −0.357*** | −0.155** | 0.135** | 0.057 | 0.167*** | 0.058 | 1.000 | | | | | |
| CTD | 0.020 | 0.034 | −0.057 | 0.002 | 0.446*** | −0.103* | −0.341*** | 1.000 | | | | |
| LTD | −0.125* | −0.098 | 0.114* | 0.100 | 0.698*** | −0.041 | 0.126* | 0.382*** | 1.000 | | | |
| OE | −0.137** | 0.120* | 0.058 | 0.098 | −0.027 | −0.029 | 0.369*** | −0.076 | −0.075 | 1.000 | | |
| logFS | −0.468*** | −0.441*** | −0.115* | −0.236*** | −0.042 | 0.140** | 0.121* | −0.168*** | −0.012 | −0.448*** | 1.000 | |
| FGR | −0.128** | −0.104* | 0.047 | −0.016 | 0.129** | 0.014 | 0.015 | 0.019 | 0.054 | −0.117* | 0.182*** | 1.000 |

Note: ROE = return on equity, $ROE_{t-1}$ = lagged return on equity, NPL = nonperforming loan, LLP = loan loss provision, LA = loan and advances, FL = financial leverage, IRR = interest rate risk, CTD = cash to total deposit ratio, LTD = loan to deposit ratio, OE = operating expense ratio, ER = efficiency ratio, FS = firm size, FGR = firm growth rate.***, **, and * indicate significance levels less than 1%, 5% and 10%, respectively.

Source: Researchers' computations.

From Table 4 below, the correlation coefficients of the lagged dependent variable, nonperforming loans, loan loss provisions, loans and advances, financial leverage, interest rate risk, cash-to-total deposit ratio, loan-to-deposit ratio, the operating expense ratio and efficiency ratio, firm size, and the firm growth rate are 60.9%, −4.9%, 0.1%, −12.8%, −22.8%, −35.7%, 2%, −12.5%, −13.7%, −46.8% and −12.8%, respectively. This result indicates a relatively strong association of lagged dependent variables, financial leverage, interest rate risk, and firm growth rate with return on assets, contrasting with other variables. The correlation coefficients between variables are all above 0.8, indicating no multicollinearity issues. However, correlation analysis lacks cumulative causes and effect relationships and reliable indicators, and econometric regression analysis is required for the main analysis.

### 3.2. Regression results for the GMM estimator

As Table 5 summarizes, the overall significances of the two models, when measured by their respective F-statistics, are 9.5 and 66.95, with P values of 0.0000, indicating that these models are well fitted at less than a 1% significance level. F-statistic results show that the model chosen for this study is appropriate and risk management indicators are associated with the performance of microfinance institutions.

As recommended by [59], to test analysis overidentification of the two-step GMM, the Hansen test P(x2) should be in the range 0.5 < P(x2) < 0.8, which is best. Option to find probability 0.1 < (x2) < 0.25. Thus, Hansen's statistics test for both models is optimal because all overidentification restrictions are valid; therefore, overidentification does not occur in either model. The other assumption test in the two-step GMM is the Arellano–Bond test of serial autocorrelation. [59] suggested the probability of Ar [2] (pr > z) shouldn't be significant at 5%. Thus, AR [2] is insignificant and greater than the 5% significance level, representing that the null hypothesis that "there is no second-order serial correlation" is not rejected. Typically, Ar [1] should be significant at 5% (AR [1] pr > z < 0.05). The null hypothesis, which says "no first-order serial correlation exists", is rejected because AR [1] is significant at less than 5% in the first differences. Accordingly, there should be no serial correlation in the original untransformed disturbances in the first differential equation.

The positive coefficient of $ROA_{t-1}$ is 0.410, which is less than 1% of significance. Similarly, the positive coefficient of $ROE_{t-1}$ is 0.169, which is below the 5% significance level and is supported by [12]. This result shows that lagged ROAt-1

**Table 5. Summary of Regression Results for The GMM Estimator.**

| Variables | ROA | ROE |
|---|---|---|
| | Coef.<br>(p-value) | Coef.<br>(p-value) |
| NPL | 0.145<br>(0.784) | −1.044<br>(0.131) |
| LLP | −1.148*<br>(0.0504) | −1.269<br>(0.127) |
| LA | −0.0196**<br>(0.0143) | 0.00724*<br>(0.0826) |
| FL | −0.00959*<br>(0.0626) | −0.0366**<br>(0.0141) |
| IRR | −0.827***<br>(0.000) | −2.043**<br>(0.0101) |
| CTD | −0.0588***<br>(0.000) | −0.112***<br>(0.000) |
| LTD | 0.00154<br>(0.830) | −0.0138<br>(0.705) |
| OE | −0.483***<br>(0.000) | −3.372***<br>(0.000) |
| ER | −0.143***<br>(0.000) | −0.342***<br>(0.000270) |
| LogFS | −0.131***<br>(0.000) | −0.406***<br>(0.000) |
| FGR | −0.00888*<br>(0.0964) | 0.00125<br>(0.951) |
| $ROA_{t-1}$ | 0.410***<br>(0.000) | |
| $ROE_{t-1}$ | | 0.169**<br>(0.0336) |
| Constant | 1.163***<br>(0.000) | 3.651***<br>(0.000) |
| F test or wald chi2 [12] | 23117.4***<br>(0.000) | 1969.98 ***<br>(0.000) |
| Observations | 240 | 240 |
| Number of C_ID | 24 | 24 |
| AR [1] Pr>z | 0.045 | 0.042 |
| AR [2] Pr>z | 0.666 | 0.255 |
| Hansen test Prob>chi2 | 0.121 | 0.422 |

Note: ROA=Return on asset, ROE=return on equity, $ROA_{t-1}$=lagged return on assets, $ROE_{t-1}$=lagged return on equity, NPL=nonperforming loan, LLP=loan loss provision, LA=loan and advances, FL=financial leverage, IRR=interest rate risk, CTD=cash to total deposit ratio, LTD=loan to deposit ratio, OE=operating expense ratio, ER=efficiency ratio, FS=firm size, FGR=firm growth rate.***, **, and * indicate the significance level less than 1%, 5% and 10%, respectively.

Source: STATA regression result.

and $ROE_{t-1}$ improve the validity of the results and indicate the risk of MFI's financial performance. The same result was found by [18,21,60]

This study investigates the association between non-performing loans (NPL), loan loss provisions (LLP), and loans and advances (LA) as indicators of credit risk and the performance of microfinance institutions (MFIs). Table 5 presents the findings. Among the credit risk indicators, NPL coefficients are 0.145 and −1.044 for ROA and ROE, respectively.

However, the p-value for LLP suggests statistical insignificance for both ROA and ROE. These results indicate an absence of a significant relationship between LLPs and MFIs performance in Ethiopia. Thus, there is no significant association between nonperforming loans and the financial performance of MFIs in Ethiopia. The negative relationship of NPL and ROE may be due to the idea that high NPLs erode shareholders' equity by increasing credit risk, impairing assets, and reducing returns on equity. Consequently, the hypothesis proposing a relationship between LLPs and MFI performance is rejected. This output is consistent with the previous study [61]. However, this finding contradicts the results of previous studies, such as those of [26] and [62].

Another indicator, Credit Risk Loans and Advances (LLP), has an estimated coefficient of −1.148, which is significant considering that ROA is below 10%. The estimated ROE coefficient is −1.269 but not significant. This non-significant negative relationship between LLP on ROE indicates the structural characteristics of Ethiopian MFIs, including reliance on external funding and their prioritization of developmental goals over shareholder returns. The same results were reported [18,22,32]. Therefore, when using ROA as a financial performance indicator, LLP exhibit a negative relationship with the financial performance of Ethiopian MFIs when measured by ROA. However, when ROE is used as the financial performance metric, no significant relationship is observed. The study accepted the hypothesis when ROA was used as the performance measure, but rejected it when ROE was used. This negative and significant relationship suggests that higher levels of LLPs lead to reduced financial performance for Ethiopian MFIs.

The third risk measure, the estimated credit and leverage coefficient (LA), are −0.0196 and −0.00724, respectively, as ROA is significant at the 5% significance level and ROE is less than 10% significant. This result shows that there is a negative relationship between LA and financial performance of MFIs. Previous researchers have reported similar results [15,20,33,34]. The proposed hypothesis is accepted. This negative and significant relationship indicates that an increase in the LA ratio leads to a decrease in the performance of MFIs.

This study uses financial leverage (FL) and interest rate risk (IRR) as market risk indicators to evaluate the performance of MFIs. Table 5 shows that, FL is negatively related to financial performance indicators ROA and ROE, with coefficients of −0.00959 and −0.0366, respectively, and is significantly less than the 10 percent level of significance with ROA and the 5 percent level of significance with ROE. The proposed hypothesis is accepted. This result is consistent with the findings of [33,34,63]. Like financial leverage, IRR is negatively related to MFIs' financial performance measures ROA and ROE, with coefficients of −0.827 and −2.043, respectively. ROA results are significant at the 1% level, while ROE results are significant at the <5% level. The proposed hypothesis is accepted. The same results were obtained by [13,15,20,32]. Financial leverage and interest rate risk are two measures of market risk; since they are negatively and significantly related to one financial performance; it shows that an increase in financial leverage and the interest rate-to-risk ratio leads to a decrease in the financial performance of MFIs.

The cash-to-total deposit ratio (CTD) and loan-to-deposit ratio (LTD) are used as liquidity risk indicators to analyze the association between liquidity risk and the financial performance of MFIs in Ethiopia. The study found that there is a negative relationship between cash-to-total deposit ratio and the financial performance of MFIs. The cash-to-total deposit ratio explains the performance of microfinance institutions with coefficients of −0.0588 and −0.112 and reaches significance of 1% of the ROA and ROE values. This negative and significant relationship indicates that an increase in the cash-to-total deposit ratio leads to a decrease in the financial performance of MFIs. The proposed hypothesis is accepted. This result is consistent with those of [12,25,36,60]. In contrast, [35] reported a positive and significant association between the cash-to-total deposit ratio and financial performance. LTD is positively correlated with financial performance indicators. The association level with ROA is insignificant, whereas it is significant for ROE. The proposed hypothesis is rejected. This result is consistent with those of [12,19,25] reported results contrary to those of this study.

The two operational risk indicators used in this study, which are the operating expense ratio (OER) and the efficiency ratio (ER), indicate a significant negative association with the financial performance of MFIs in Ethiopia. The OER explains the financial performance of MFIs with coefficients of −0.483 and −3.372. This result is significant at the 1 percent

significance level for both ROA and ROE. Thus, the proposed hypothesis is accepted. A similar finding reported by [10]. OER and ER explain the financial performance of MFIs with coefficients of −0.143 and −0.342, respectively. OER and ER are statistically significant at the 1 percent significance level for both ROA and ROE. Thus, the proposed hypothesis is accepted. This result is similar with the finding of [10]. This significant and negative association between the operating expense ratio and the efficiency ratio, both of which serve as measures of operational risk, suggests that an increase in financial leverage and the interest rate-to-risk ratio results in a decline in the financial performance of MFIs.

In general, in this study, the regression results of the variable signs and levels of significance are not the same for either of the two financial performance indicators. This is because financial performance indicators do not indicate the financial performance of MFIs equally because they use different formulas. For example, return on assets indicates the overall efficiency of management and reflects whether MFIs use assets and liabilities effectively to produce their income. The return on equity provides information on how well managers use the funds that shareholders invest without considering the firm's liability effect.

## 4. Conclusions and recommendations and implications

The study found that lagged dependent variables ROA and ROE are positively associated with financial performance, while NPL, LLP and LA have a negative association. Financial leverage and interest rate risk are negatively associated with ROA and ROE. Liquidity risk indicators CTD and LDRs show a negative association with financial performance while operating expense and efficiency ratios show a negative association.

Attention should be given to financial leverage, interest rate risk, and cash-to-total deposits since they negatively affect the ROA and ROE of MFIs in Ethiopia. Financial leverage is all about managing debt in a way that enhances profitability while keeping risks under control. MFIs in Ethiopia should focus on maintaining an optimal debt-to-equity ratio to avoid excessive leverage that could lead to financial distress. They should consider a balanced approach to borrowing, ensuring that the cost of debt is lower than the return on assets. Thus, MFIs in Ethiopia should conduct a thorough analysis of their debt structure and its impact on ROA and ROE and implement effective risk management strategies to mitigate the negative effects of high leverage. To mitigate interest risk, MFIs in Ethiopia should focus on implementing effective interest rate risk management policies and strategies. Thus, MFIs in Ethiopia should regularly monitor interest rate movements and their impact on profitability. Maintaining an appropriate level of the cash-to-total deposit ratio is crucial for ensuring liquidity and operational efficiency. MFIs in Ethiopia should focus on a balance between liquidity and profitability to optimize their financial performance. Generally, to mitigate the effects of financial leverage, interest rate risk, and cash-to-total deposits on ROA and ROE, MFIs in Ethiopia should focus on implementing effective risk management policies and strategies.

Since loan loss provisions have a significant negative association with ROA, enhancing the loan loss provisioning practices of MFIs in Ethiopia is crucial for mitigating the impact of non-performing loans on profitability. Given the association between loans and advances and financial performance, MFIs should focus on optimizing their loan portfolio to improve ROA by regularly monitoring loan performance metrics to identify potential risks and opportunities for improvement.

Given the positive association between the loan-to-deposit ratio and financial performance indicators, MFIs in Ethiopia should focus on optimizing their loan-to-deposit ratio to maximize profitability while maintaining liquidity by regularly reviewing and adjusting the loan-to-deposit ratio to align with the business goals and market conditions of MIFs. Although there is no significant association between nonperforming loans and financial performance, it is still crucial for MFIs to prioritize the effective management of nonperforming loans to maintain asset quality and borrower confidence. Generally, this study provides valuable insights for enhancing the performance and sustainability of Ethiopian MFIs, with implications for broader contexts. Managers can use these insights to refine lending practices, strengthen credit evaluation, and better align financial strategies with business goals. Policymakers should foster an enabling environment through financial literacy initiatives, risk management frameworks, and regulatory support, while MFI managers can focus on leveraging historical performance, optimizing liquidity, and strengthening credit evaluation. The study also highlights the relevance of

its findings beyond Ethiopia, suggesting that other countries with similar financial ecosystems can adapt these insights for context-specific improvements. Additionally, the study provides investors with key insights to assess MFI sustainability by identifying performance-related indicators like leverage, liquidity, and credit risk, aiding informed decisions and improving risk monitoring frameworks. Although this study is specific to Ethiopia, the findings are relevant to similar financial ecosystems globally, highlighting the need for context-specific adaptations. Regarding further research directions, future research could improve this by expanding the sample size since the power test result is below the convectional threshold, indicating that the cross-sectional component of this study may have a reduced ability to detect moderate relationships between the independent and dependent variables, focusing on manufacturing or merchandising businesses, and using long-term series data for better insights. Although this study did not examine technological advancements and financial innovations, such as mobile banking and financial technology solutions, these tools are increasingly vital for improving risk management, operational efficiency, and financial performance in MFIs. In addition, future research should explore how Ethiopian MFIs adopt these technologies and their impact, as integrating such variables could offer a more comprehensive understanding of sector performance. Policymakers and practitioners should also prioritize digital transformation in the microfinance landscape. Moreover, Future research should examine how digital tools, regulatory flexibility, financial literacy, and diverse funding models impact the efficiency, sustainability, and outreach of Ethiopian MFIs, especially in rural areas.

## Supporting information

**S1 File. Data.**
(XLSX)

## Author contributions

**Conceptualization:** Megbaru Tesfaw Molla, Ratinder Kaur.

**Data curation:** Megbaru Tesfaw Molla.

**Formal analysis:** Megbaru Tesfaw Molla, Ratinder Kaur.

**Investigation:** Megbaru Tesfaw Molla, Ratinder Kaur.

**Methodology:** Megbaru Tesfaw Molla, Ratinder Kaur.

**Project administration:** Megbaru Tesfaw Molla.

**Resources:** Megbaru Tesfaw Molla.

**Software:** Megbaru Tesfaw Molla, Ratinder Kaur.

**Supervision:** Megbaru Tesfaw Molla.

**Validation:** Megbaru Tesfaw Molla, Ratinder Kaur.

**Visualization:** Megbaru Tesfaw Molla.

**Writing – original draft:** Megbaru Tesfaw Molla.

**Writing – review & editing:** Megbaru Tesfaw Molla.

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
