## [Decision Letter · Decision Letter 0]

PONE-D-24-16639ASSOCIATION OF RISK MANAGEMENT PRACTICES AND FINANCIAL PERFORMANCE OF MICROFINANCE INSTITUTIONS IN ETHIOPIAPLOS ONE

Dear Dr. Molla,

Thank you for submitting your manuscript to PLOS ONE. After careful consideration, we feel that it has merit but does not fully meet PLOS ONE’s publication criteria as it currently stands. Therefore, we invite you to submit a revised version of the manuscript that addresses the points raised during the review process.

The manuscript has been evaluated by three reviewers, and their comments are available below.

The reviewers have raised a number of major concerns. Most importantly, Reviewer 3 points out the need for greater methodological rigour, especially in justifying the GMM estimator and the sample size, and also the need to ensure the validity and reliability of the secondary data. Please note, however, that we do not require that the study be expanded to other countries or to include the impact on clients, as requested by Reviewer 1, and we do not feel that the citation suggestions of Reviewer 2 are appropriate. 

We therefore ask you to revise the manuscript in response to the reviewer requests and the editorial comments above.

We look forward to receiving your revised manuscript.

Kind regards,

Patrick Goymer

Staff Editor

PLOS ONE

Journal Requirements:

“The authors assert that they do not have any conflicting interests”

4. In the online submission form, you indicated that [Data is available in the hand of authors. Upon request by the Journal, the authors can submit it.]. 

Reviewers' comments:

Reviewer's Responses to Questions

**Comments to the Author**

1. Is the manuscript technically sound, and do the data support the conclusions?

Reviewer #1: Partly

Reviewer #2: Yes

Reviewer #3: No

2. Has the statistical analysis been performed appropriately and rigorously? 

Reviewer #1: Yes

Reviewer #2: Yes

Reviewer #3: No

3. Have the authors made all data underlying the findings in their manuscript fully available?

Reviewer #1: No

Reviewer #2: Yes

Reviewer #3: No

4. Is the manuscript presented in an intelligible fashion and written in standard English?

Reviewer #1: Yes

Reviewer #2: Yes

Reviewer #3: No

5. Review Comments to the Author

Reviewer #1: Referee report for

PONE-D-24-16639

TITLE: ASSOCIATION OF RISK MANAGEMENT PRACTICES AND FINANCIAL PERFORMANCE OF MICROFINANCE INSTITUTIONS IN ETHIOPIA

To the Authors

This paper analyzes how Microfinance institutions provide financial services to low-income clients, especially rural women, and have grown significantly to compete with formal banks. The authors exploit a change in legislation promoted by Ethiopia's National Bank occurred in 1996, to evaluate the relationship between risk management practices and the financial performance of 24 Ethiopian MFIs from 2013 to 2023.

Compared to the existing literature, the authors offer a comprehensive view of the risks affecting MFIs’ management practices, over a relatively extended period, and integrating the regulatory context in the risk analysis.

I found that there are some notable contributions and potential new aspects of this research:

1. Comprehensive Risk Management Analysis. The study examines a broad range of risks affecting MFIs’ management practices. Prior research often focused predominantly on credit risk. Hence, including a wider spectrum of (strategic, credit, liquidity, interest rates, and operational) risks provides a more holistic view of the factors influencing MFI performance.

2. Regulatory Context: The study considers the impact of Ethiopia's microfinance legislation introduced in 1996 and subsequent regulatory developments. Understanding how regulatory changes affect MFIs' risk management and performance is a relatively novel aspect, as most studies do not integrate the regulatory dimension deeply.

3. Methodological Approach. By challenging the mixed and inconsistent findings of previous research on the relationship between risk management and MFIs’ performance, the methodology and the specific analytical techniques used in this study could offer new perspectives or validate findings with stronger empirical evidence.

***

MAIN CONCERNS

Below are some potential areas that might be missing or could benefit from further exploration:

1. Comparative Analysis with Other Regions:

While the focus on Ethiopian MFIs provides important insights, a comparative analysis with MFIs in other countries or regions could offer a broader perspective and help identify unique or common challenges and best practices.

2. Client Impact Assessment:

The study focuses on the financial performance of MFIs, but it might not thoroughly assess the impact of MFIs on their clients, especially the socio-economic outcomes for low-income households and women. Evaluating (or at least discussing) the direct benefits to clients could provide a more comprehensive view of MFI effectiveness.

4. Technology and Innovation in MFIs:

The role of technological advancements and financial innovations (e.g., mobile banking, fintech solutions) in improving risk management and financial performance is not taken into consideration. Exploring/discussing how these technologies are integrated into Ethiopian MFIs could be valuable.

5. Detailed Examination of External Factors:

The study should benefit from fully accounting for external factors such as economic conditions, political stability, and social dynamics that can influence MFIs’ performance and risk management practices. A more detailed analysis and inclusion of these external influences could enhance the study's findings. Furthermore, assessing environmental, social (e.g., ESG), and governance factors could provide a more comprehensive view of MFI performance. Finally, the influence of local cultural and behavioral factors on risk management practices and MFI performance should be thoroughly examined.

***

OTHER CONCERNS

• Data Period and Sample Size. The paper uses data collected from 24 MFIs over a decade (2013 to 23). This extended period might offer robust and reliable insights in the medium run, compared to other studies focusing on shorter time frames. However, the sample size is very small (24 MFIs); I suggest adding power tests.

• Practical Implications for Policy and Practice. The study’s findings could have significant implications for policymakers, MFI managers, and stakeholders, although confined to Ethiopia. I suggest discussing more in-depth the external validity of the paper’s findings, clearly indicating what can enhance the financial performance and sustainability of MFIs in other contexts.

Reviewer #2: 1. Title and theme of the paper is justified. But title needs revision. Make a scientific title following the call for paper of this journal.

2. Overall structure of thematic content is OK. In my opinion, authors have taken a limited approach to measure and study cleaner environment. Authors must take a multidimensional approach to study natural resources and green economic recovery that need to be improved.

3. The abstract of the manuscript is defined and precise but still there is a room to improve it. However, it should discuss the finding of the study along with policy implications in a way that reader generates interest to read the whole paper. If possible, please add 1-2 lines about policy implications in the abstract.

Following citation may enhance your literature review

https://www.sciencedirect.com/science/article/abs/pii/S0301421520307631

https://doi.org/10.1016/j.techfore.2023.122872

https://doi.org/10.1016/j.resourpol.2023.103614

https://doi.org/10.1016/j.eneco.2023.106749

https://doi.org/10.1016/j.rser.2023.113321

https://doi.org/10.1016/j.resourpol.2023.103508

https://doi.org/10.1016/j.envdev.2022.100794

https://doi.org/10.1016/j.resourpol.2023.103780

https://doi.org/10.1016/j.resourpol.2023.104216

4. Introduction section requires to explain the novelty of the paper including study motivation, contribution and research problem. Please try to revise the contribution and study motivation paragraph with brief and comprehensive detail.

5. The contribution seems dim in this manuscript. I would suggest the author to enhance your theoretical discussion and arrives your debate or argument to show satisfactory contribution. Refine research contribution and objectives. Clearly state both theoretical and practical contributions.

6. The authors have failed to explain the gap for the current study, although on page 6 they have mentioned a brief summary, but just naming some techniques is not sufficient to highlight the importance of current study. Please revise.

7. Literature review section is finely attempted but this section still need improvement. I think the authors must revisit literature again and re-write in fine academic way. Because as in it's present state it is not sufficient.

8. Overall, the methodology of study looks fine,

Rectify syntax errors and grammatical issues in it. Particularly, you can improve the advantages of this and its application with broader context.

10. Overall, results section is also fine. Add rigor in this section and present some graphs in this section. Add some more options to develop rigor in results and discussion section. In results sections, link your study results with the previous studies. Give a comparison of findings in your results study. Make a sequential orders of all results sections also cite table numbers in it. Add some beautiful graphs too.

11. Suitable captions and heading title of the figures and tables are needed before acceptance of the paper.

13. The policy response of different countries should be explained in detail along with the policy recommendations.

Reviewer #3: Review Comments for the Article: "ASSOCIATION OF RISK MANAGEMENT PRACTICES AND FINANCIAL PERFORMANCE OF MICROFINANCE INSTITUTIONS IN ETHIOPIA"

1. Research Relevance and Contribution: While the topic of risk management practices and financial performance of microfinance institutions (MFIs) is of importance, the article does not sufficiently clarify its contribution to the existing literature. It lacks a thorough review of previous studies to establish the novelty of the research and its implications for policymakers and practitioners.

2. Methodology Issues: The study employs a mixed research approach; however, further details are necessary to understand how qualitative data were integrated with quantitative findings. The methodological rigor appears to be inadequate due to the absence of clear justification for the selection of the GMM estimator and the sample size. The decision-making process regarding the selection of the 24 MFIs should be better explained to enhance reproducibility.

3. Data Collection and Analysis: The article mentions secondary data collection from financial statements, but it does not discuss the methods used to ensure the reliability and validity of this data. Additionally, while the statistical analysis with STATA is noted, there is insufficient detail on how the analysis controlled for potential confounding variables and biases.

4. Findings and Interpretations: The findings highlight important relationships, but there is a lack of depth in the interpretation of these results. For instance, the report of negative relationships should explore potential reasons behind these phenomena. Also, the implications of non-significant relationships, particularly concerning nonperforming loans and performance indicators, warrant further exploration.

5. Presentation and Clarity: The writing lacks clarity in some sections, making it challenging to follow the argumentation. More precise language and structure are needed to improve comprehension. Additionally, several technical terms are used without adequate explanation, which could alienate readers unfamiliar with the subject matter.

6. Conclusion and Recommendations: The conclusion seems to overgeneralize the findings without providing actionable recommendations. It would enhance the article to include suggestions for practitioners in the microfinance sector based on the study's results.

7. References and Citations: The references cited in the article are not extensive and seem outdated in some areas. An updated literature review would bolster the research's credibility and context.

Decision: Based on the aforementioned points, I regret to inform you that I cannot recommend the publication of this article in its current form. I encourage the authors to address the highlighted areas of concern before resubmitting for consideration in a suitable forum.

6. PLOS authors have the option to publish the peer review history of their article (what does this mean? ). If published, this will include your full peer review and any attached files.

**Do you want your identity to be public for this peer review?** For information about this choice, including consent withdrawal, please see our Privacy Policy .

Reviewer #1: No

Reviewer #2: No

Reviewer #3: No

---

## [Author Response · Author response to Decision Letter 1]

24 Dec 2024

Thank you for the opportunity to revise our manuscript titled “ASSOCIATION OF RISK MANAGEMENT PRACTICES AND FINANCIAL PERFORMANCE OF MICROFINANCE INSTITUTIONS IN ETHIOPIA” and for the thoughtful comments and suggestions provided by the reviewers. We greatly appreciate your time and also, we thank the reviewers for their time and effort, as the feedbacks have helped to improve the clarity, depth, and overall quality of the paper. Below, we provide a detailed point-by-point response to the comments raised. For clarity, comments are presented in bold, followed by our responses in plain text. All changes made in the manuscript are highlighted in red color. We trust that the revisions and clarifications made in response to the comments have strengthened the manuscript. If you have any further questions or require additional modifications, we would be happy to address them promptly.

---

## [Decision Letter · Decision Letter 1]

PONE-D-24-16639R1ASSOCIATION OF RISK MANAGEMENT PRACTICES AND FINANCIAL PERFORMANCE OF MICROFINANCE INSTITUTIONS IN ETHIOPIAPLOS ONE

Dear Dr. Molla,

Thank you for submitting your revised manuscript to PLOS ONE. After careful consideration, we feel that while it has merit and substantial improvements have been made it does not fully meet PLOS ONE’s publication criteria as it currently stands. Therefore, we invite you to submit a revised version of the manuscript that addresses the points raised during the review process.

We look forward to receiving your revised manuscript.

Kind regards,

Kennedy Munyua Waweru, PhD

Academic Editor

PLOS ONE

Journal Requirements:

Additional Editor Comments:

>Please note that the refined suggestions of the R1 relating to commenting and providing a reflection on comparative analysis other countries or regions and the role of technological advancements and financial innovations should be addressed since they just require do be drawn from literature review- No expansion of scope of data collection is required or necessary.

>Power tests are also necessary

>Please however NOTE that while review of current relevant and literature is a scholarly expectation, we DO NOT require you to cite the references given by reviewer 4.

Reviewers' comments:

Reviewer's Responses to Questions

**Comments to the Author**

1. If the authors have adequately addressed your comments raised in a previous round of review and you feel that this manuscript is now acceptable for publication, you may indicate that here to bypass the “Comments to the Author” section, enter your conflict of interest statement in the “Confidential to Editor” section, and submit your "Accept" recommendation.

Reviewer #1: (No Response)

Reviewer #4: All comments have been addressed

2. Is the manuscript technically sound, and do the data support the conclusions?

Reviewer #1: Partly

Reviewer #4: Yes

3. Has the statistical analysis been performed appropriately and rigorously? 

Reviewer #1: Yes

Reviewer #4: Yes

4. Have the authors made all data underlying the findings in their manuscript fully available?

Reviewer #1: Yes

Reviewer #4: Yes

5. Is the manuscript presented in an intelligible fashion and written in standard English?

Reviewer #1: Yes

Reviewer #4: Yes

6. Review Comments to the Author

Reviewer #1: Referee report for

PONE-D-24-16639_R1

TITLE: ASSOCIATION OF RISK MANAGEMENT PRACTICES AND FINANCIAL PERFORMANCE OF MICROFINANCE INSTITUTIONS IN ETHIOPIA

To the Authors

Unfortunately, few or none of the issues I raised in my previous report were taken into consideration, some did not even require a particular effort in terms of information and data collection. Therefore, I do not understand the preponderance of rebuttals on the issues I raised.

As highlighted in my first report there are some notable contributions and potential new aspects of this research that could make the paper more interesting for a wider audience and not confined to the specific issues of Ethiopia. This is a fundamental aspect that I believe cannot be bypassed.

In particular:

“Comparative analysis with MFIs in other countries or regions could offer a broader perspective and help identify unique or common challenges and best practices”.

=> The authors provide a rebuttal that does not make much sense. Based on their experience in microfinance, and their knowledge of the specific context of Ethiopia, they could provide useful indications in terms of exportability of the results obtained in other developing countries that have similar institutional conditions to Ethiopia. This can be commented on without necessarily resorting to further data collection. However, if this were necessary, one could rely on synthetic and aggregate indicators developed by platforms such as MIX Market, with minimal effort in terms of processing and comparison.

“The role of technological advancements and financial innovations (e.g., mobile banking, fintech solutions) in improving risk management and financial performance is not taken into consideration. Exploring/discussing how these technologies are integrated into Ethiopian MFIs could be valuable.”

=> The authors argue that they regret to expand the scope of this study to technological and innovation in MFIs without even providing an argument in this regard. I ask them not to underestimate this aspect, at least try to provide some reflections in the conclusions.

“Data Period and Sample Size. The paper uses data collected from 24 MFIs over a decade (2013 to 23). This extended period might offer robust and reliable insights in the medium run, compared to other studies focusing on shorter time frames. However, the sample size is very small (24 MFIs); I suggest adding power tests.”

=> I understand that the authors are constrained by the availability of data on 24 MFIs only. But avoiding providing power tests is not justifiable.

Reviewer #4: Abstract:

This manuscript manages to provide a well summary of the vital essence. The issue of risk management and financial performance of microfinance Institutions (MFIs) in Ethopia is interesting to discuss. However, several prior studies have focused on similar issues but to other different samples of countries, therefore this study should significantly highlight the novelty of this paper’s issue. The GMM estimation methods have been mentioned in the abstract. The results obtained are also summarized.

Introduction:

In general, this study highlighted the issue specifically on the MFIs specifically on the perception of risk management and financial performance . The discussions are well, and the details explained in the introduction provide a general view of these issues to the readers

Literature review:

1) The literature review has comprehensively structured by summarizing all the prior studies on the issue discussed.

2) This manuscript demonstrates an adequate understanding of the relevant literature in the field and cite an appropriate range of literature sources

3) All the significant work are taken into account. The correlation between literature review and results, the correlation between objectives are clearly stated

4) The citations of previous studies on financial institution, banking sectors, MFI efficiency, productivity, and performance must be included, as the issues of risk and financial performance are interrelated but currently underrepresented. To enhance the study's contribution, the author(s) should incorporate more relevant studies that address these issues, along with updated research articles. It is needed to cite the following nine prior articles in the literature review.:

1) Price Efficiency and Returns to Scale of Banking Sector in Gulf Cooperative Council Countries: Empirical Evidence from Islamic and Conventional Banks (2013). Economic Computation and Economic Cybernetics Studies and Research.

https://ecocyb.ase.ro/nr.3.pdf/Fakarudin%20Kamarudina.pdf

2) Institutional factors and efficiency performance in the global microfinance industry (2023). Benchmarking: An International Journal

https://doi.org/10.1108/bij-06-2021-0326

3) A DEA and random forest regression approach to studying bank efficiency and corporate governance (2022). Journal of the Operational Research Society

https://doi.org/10.1080/01605682.2021.1907239

4) Does country governance and bank productivity Nexus matters? (2022). Journal of Islamic Marketing

https://doi.org/10.1108/jima-05-2019-0109

5) Revisiting efficiency of microfinance institutions (MFIs): an application of network data envelopment analysis (2021). Economic Research-Ekonomska Istraživanja

https://doi.org/10.1080/1331677x.2020.1819853

6) COVID-19 crisis and the efficiency of Indian banks: Have they weathered the storm? (2023). Socio-Economic Planning Sciences

https://doi.org/10.1016/j.seps.2023.101661

7) Unboxing the black box on the dimensions of social globalisation and the efficiency of microfinance institutions in Asia (2021). Oeconomia Copernicana

https://doi.org/10.24136/oc.2021.019

8) The impact of ownership structure on bank productivity and efficiency: Evidence from semi-parametric Malmquist Productivity Index (2014). Cogent Economics and Finance

https://doi.org/10.1080/23322039.2014.932700

9) Islamic Banking Sectors in Gulf Cooperative Council Countries: Analysis on Revenue, Cost and Profit Efficiency Concepts (2014). Journal Of Economic Cooperation and Development.

https://jecd.sesric.org/pdf.php?file=ART12092401-2.pdf

Therefore, by incorporating and citing these studies in the literature review, the gaps in existing research will be clearly identified, and the unique contribution of this study will be effectively highlighted.

Methodology:

1) The data sample of the 24 MFIs (10 years) are adequate

2) Relevant estimation method used in the analysis

Results & discussion:

1) Results are clearly presented and analyzed appropriately

2) Excellent and interesting

3) Detail justification has been provided by the author(s)

Conclusion:

1) Well summarized, including all the vital points.

2) However, the author(s) need to explain more on the implication of this study on how the results benefiting by the related parties

7. PLOS authors have the option to publish the peer review history of their article (what does this mean? ). If published, this will include your full peer review and any attached files.

**Do you want your identity to be public for this peer review?** For information about this choice, including consent withdrawal, please see our Privacy Policy .

Reviewer #1: No

Reviewer #4: No

---

## [Author Response · Author response to Decision Letter 2]

12 May 2025

we thank you the reviewer for your important issues raised.

---

## [Decision Letter · Decision Letter 2]

Association of Risk Management Practices and Financial Performance of Microfinance Institutions in Ethiopia: A Two-Step System Generalized Method of Moments Approach

PONE-D-24-16639R2

Dear Dr. Molla,

We’re pleased to inform you that your manuscript has been judged scientifically suitable for publication and will be formally accepted for publication once it meets all outstanding technical requirements.

Kind regards,

Kennedy Munyua Waweru, PhD

Academic Editor

PLOS ONE

Additional Editor Comments (optional):

Reviewers' comments:

Reviewer's Responses to Questions

**Comments to the Author**

1. If the authors have adequately addressed your comments raised in a previous round of review and you feel that this manuscript is now acceptable for publication, you may indicate that here to bypass the “Comments to the Author” section, enter your conflict of interest statement in the “Confidential to Editor” section, and submit your "Accept" recommendation.

Reviewer #1: All comments have been addressed

Reviewer #4: (No Response)

2. Is the manuscript technically sound, and do the data support the conclusions?

Reviewer #1: Yes

Reviewer #4: Partly

3. Has the statistical analysis been performed appropriately and rigorously? 

Reviewer #1: Yes

Reviewer #4: N/A

4. Have the authors made all data underlying the findings in their manuscript fully available?

Reviewer #1: Yes

Reviewer #4: No

5. Is the manuscript presented in an intelligible fashion and written in standard English?

Reviewer #1: Yes

Reviewer #4: No

6. Review Comments to the Author

Reviewer #1: The authors have addressed all the comments raised in the previous round of review in a clear and satisfactory manner. The revised manuscript now presents a well-articulated research question, a more robust theoretical framework, and a clearly described methodology.

Reviewer #4: I am disappointed that the authors did not appear to take the reviewers' comments seriously, as the revised manuscript shows little to no meaningful improvement. Given the lack of sufficient revision and engagement with the feedback provided, I have no further comments and recommend that the manuscript be rejected for publication.

7. PLOS authors have the option to publish the peer review history of their article (what does this mean? ). If published, this will include your full peer review and any attached files.

**Do you want your identity to be public for this peer review?** For information about this choice, including consent withdrawal, please see our Privacy Policy .

Reviewer #1: **Yes: ** LUCIA Dalla PELLEGRINA

Reviewer #4: No

---

## [Editor Report · Acceptance letter]

PONE-D-24-16639R2

PLOS ONE

Dear Dr. Molla,

I'm pleased to inform you that your manuscript has been deemed suitable for publication in PLOS ONE. Congratulations! Your manuscript is now being handed over to our production team.

Kind regards,

on behalf of

Professor Kennedy Munyua Waweru

Academic Editor

PLOS ONE